# Hydrology and Cranes (*Grus grus*) Attraction Partnership in the Management of the Hula Valley—Lake Kinneret Landscape

**Moshe Gophen**

MIGAL Scientific Research Institute, P.O. Box 831, Kiryat Shmone 11016, Israel; Gophen@Migal.org.il

**Abstract:** The Hula Valley in northern Israel was partly covered by swamps and a shallow lake. The entire valley was drained and converted for agricultural cultivation. Later, an additional soil reclamation operation was implemented, including eco-tourism. From the early 1990s, winter migratory cranes have attracted visitors, thus supporting the hydrological management of the entire valley that protects the downstream Lake Kinneret. It was documented that these birds have a minor impact on phosphorus pollution, but severely damaged agricultural crops are protected by mild deportation and daily, short, periodical corn seed feeding.

**Keywords:** cranes; deportation; crop protection; Hula Valley; phosphorus; pollution

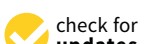



## 1. Introduction

### 1.1. The Kinneret Drainage Basin

The region that is geographically defined as the Hula Valley (altitude between 100 and 60 mbsl) has a surface area of about 200 km$^2$ and provides about 7% of the watershed basin of Lake Kinneret [1] (Figure 1). The historic Hula wetland area within the valley generally refers to about 60 km$^2$ of the entire valley [1–3]. The regional climate conditions are Mediterranean [1], with an annual winter rainfall of 350–850 mm, followed by a hot and dry summer. The temperature minima and maxima are, respectively, 18 and 35 °C in summer and 3 and 20 °C in winter [3,4]. The Hula Valley and the surrounding slopes are one of the most ancient human habitats [5], dating back to 73,000 BC.

The Kinneret drainage basin (2730 km$^2$, altitude range 2284–2260 mbsl) is located mostly to the north of the lake. Its maximum length from north to south is 110 km, and the width is 50 km [6]. During the past 80 years, the Lake Kinneret drainage basin ecosystem has undergone significant anthropogenic and natural modifications. Prior to the 1950s, the Hula Valley was mostly (6500 ha) covered with old Lake Hula (1300 ha) and swampy wetlands. This area was not cultivated, malaria was common, and water loss by evapo-transpiration (ET) was high. The Jordan River, which crosses the Hula Valley, contributes about 63% of the downstream Lake Kinneret's water budget and 70% of the total nutrient inputs, of which over 50% originate from the Hula Valley region, including the valley and the slopes on both sides (east and west) of it. The old Lake Hula and its swamps were drained and converted for agricultural development. After the Hula drainage, the dominant N flux from the peat organic soil was modified from ammonium to nitrate. As a result of raw sewage removal and fishpond restrictions, the organic nitrogen flux into Lake Kinneret significantly reduced. As a result of inappropriate irrigation and agricultural methods, the peat soil quality deteriorated due to consolidation, destruction, and surface subsidence. This deterioration was accompanied by heavy dust storms [6] blocking drainage canals, due to peat organic soil dryness, resulting in deep cracks. This was followed by oxygen penetration, which accompanied exogenic decomposition of organic matter and heating, creating underground fires. The underground cracks, together with cultivated plant mater, immediately were populated by severe outbreaks of rodent populations. These deterioration processes caused severe damage to agricultural crops.

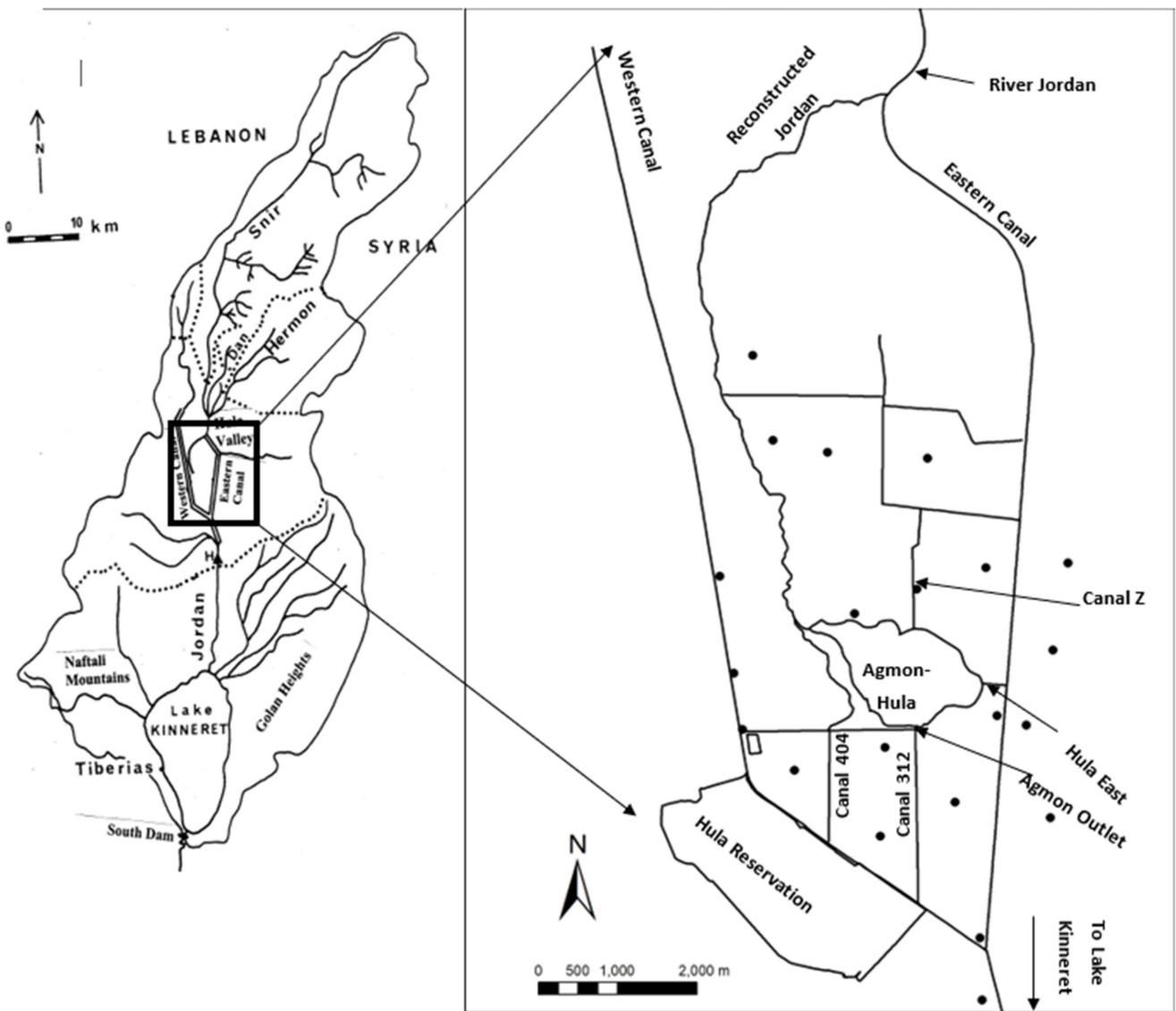

**Figure 1.** Geographical maps of the Lake Kinneret watershed with inflow rives (**left**) and the Hula Valley (**right**). Bore holes drills are indicated (black spots).

### 1.2. The Hula Reclamation Project

The Hula Reclamation Project (HRP) (Figure 1) was consequently implemented. The implementation of the HRP resulted in modifications to land use, which affected the regime of nutrient supply to Lake Kinneret. The HRP included the creation of a new shallow lake named Agmon (surface area 820 ha, mean depth 0.2 m, volume $0.164 \times 10^6$ m$^3$) [7]; the renewal of 90 km drainage and water supply canals, placing a vertical plastic barrier of depth 4.5 m such that it spanned a distance of 2.8 km along the valley from east to west; maintenance of a higher underground water table; and functional conversion of 500 ha of the Hula Valley land, including Lake Agmon in the center, from agricultural to eco-tourism usage. The objectives of the HRP were (1) nutrient removal from the Lake Kinneret external loads through the Lake Agmon hydrological system; (2) production of an ecological component for eco-tourism, i.e., Lake Agmon; (3) usage of Lake Agmon as a principle component of hydrological management and agricultural irrigation for the entire valley; (4) improvement of irrigation water supply through portable computerized spray water lines; (5) maintenance of a high underground water table to enhance peat soil moisture and prevent peat soil quality deterioration; and (6) re-establishment of highly diverse natural flora and fauna, with emphasis on aquatic birds.

### 1.3. Cranes in the Hula Valley

In the early 1990s, peanut cultivation became attractive due to its high beneficial value and productive success as a result of the suitability of the peat organic soil. The winter migrators, cranes (*Grus grus*), on their long-term southern traditional migration to Africa, discovered some leftover peanuts after harvest [8,9]. Thus, the flocks massively landed in October–November in the Hula Valley. The population of cranes in the Hula Valley therefore increased. Unfortunately, when the rains came, the wet peanuts started to ferment and became less attractive to the cranes, resulting in their looking for other food sources, thereby causing damage to agricultural crops. At this point, a conflict arose: on the one hand, cranes attracted bird watchers (eco-tourism), but on the other hand, crop damage became common, where significant water supply enhanced agricultural development, which was damaged by the birds [10,11]. The rationale behind the Hula Reclamation Project was the protection of the Lake Kinneret water quality, combined with the production of beneficial (income source) crops for the citizens of northern Israel. The advantages of the Hula Reclamation Project included additional water allocation and improvement of the canal and irrigation systems (portable computerized spray lines) to improve crops, as well as development of eco-tourism where the soil was unsuitable for agriculture [7,12]. The increase in the crane population resulted in damaging these benefits.

This paper is a long-term (1994–2018) summary aimed at evaluating the combined effects of the Hula Reclamation Project achievements on Lakes Agmon and Kinneret and attempts at forecasting future beneficial management design for all partners involved: farmers, water managers, and nature protectors. The paper is focused on a controversial issue of the potential impact of cranes on the pollution of Lake Kinneret. The Hula Valley comprised only about 7% of the total watershed of Lake Kinnerer (2730 km$^2$). Moreover, the size of the cultivated part, together with the eco-touristic block (including Lake Agmon) (the peat soil land) is just 3% of the total drainage basin area. The significant role of the peat soil in nitrogen contribution was documented. Nevertheless, the involvement of the peat soil phosphorus within the Lake Kinneret input load dynamics was just partly studied. Peat soil dust wind release and deposit onto Lake Kinneret were documented [6]. The phosphorus capacity within the Hula Valley runoffs as well as their inflows into River Jordan were also documented. Nevertheless, partitioning of the phosphorus stock sources, other fertilizers, crane droppings, and geochemical or erosion processes has not yet been studied. This paper is an attempt at estimating the role of cranes within the entire load. Because the peat soil dust contribution, not like other sources, was estimated, the paper is focused on the relative potential crane impact within the comprehensive phosphorus delivery from the Hula Valley into River Jordan. The research about phosphorus dynamics within the Lake Kinneret drainage basin initiates a significant impact of climate change on it. Therefore, re-evaluation of the climate condition was carried out aimed at the search for the domination of environmental parameters: edaphic, hydrological, climate, agriculture, and crane migration. The particular issue that this paper intends to emphasize is that the reason for an increase in the TP concentration in the Lake Agmon waters is submerged macrophytes and not crane droppings.

## 2. Materials and Methods

Temporal changes in the Lake Kinneret drainage basin of precipitation and River Jordan discharge (climate change) conditions are given in Figure 1. The experimental methods and sampling program have been documented and published earlier [13–19]. A brief, summarized compilation of previously collected information is re-evaluated here. Data sources and statistical methods are given below.

### 2.1. Data Sources

The long-term datasets (1970–2018) of Lake Kinneret and its watershed, including data on the water and precipitation, nutrient dynamics, and river discharge, were statistically evaluated. Data were obtained from the following sources: the Kinneret Limnological

Laboratory [13–19], annual reports of the Israeli National Meteorological Service, the Israeli National Hydrological Service (National Water Authority), MIGAL, the Hula Project Service, the Mekorot Water Supply Company Ltd., the Monitoring Unit Jordan District, and the Agriculture Ministry Northern Branch—Upper Galilee Office.

*2.2. Statistical Method*

Statistical analyses by fractional polynomial regression, linear regression with 95% confidence intervals, simple averages, and line scatter plots were carried out using STATA 9.1.

These are the abbreviations used (Table 1):

TPin—total phosphorus concentration (ppb, ppm) or mass load (ton) in Lake Agmon inflow sources
TPout—total phosphorus concentration (ppb, ppm) or mass load (ton) in Lake Agmon outflow
TPbalance—mass balance (ton) = outflow minus inflow.

## 3. Results

Information about climate change [4,20] in the Hula–Lake Kinneret region is given in Figure 1. There was a significant decline in precipitation and a consequent reduction in the River Jordan mean discharge by 454,106 m$^3$/year and 13,510$^6$ m$^3$/year during the periods of 1970–2001 and 2001–2018, respectively (Figure 2). The decline in discharge was followed by a significant decrease in nutrient input into Lake Kinneret and a lowered water level (WL) in the Hula Valley. Results presented in Figure 2 indicate that not as documented in many systems, the relationship between discharge and the total phosphorus (TP) concentration is positive: the higher the discharge, the higher the TP concentration [21–23]. Climate conditions in the Hula–Lake Kinneret region is defined as tropical, with a short, cold and wet winter and a long, warm, and dry summer. Consequently, the River Jordan discharge is high in winter and low in summer (Figure 2). The temporal decline in the TP concentration in the River Jordan waters with respect to a decline in discharge is shown in Figure 3 by two statistical methods: linear regression (95% CI) and LOWESS smoothing (bandwidth = 0.8). Similar temporal trends are shown in Figure 4 for changes in the concentrations of organic nitrogen and total nitrogen. Figure 5 gives the inverse relationship between TP annual mean concentrations in the Lake Agmon outflow and in the River Jordan waters. It is presented by two statistical methods: linear regression with 95% confidence intervals and fractional polynomial regression. A prominent indication of TP concentration decline in Lake Agmon waters is correlated with an increase in TP concentration in the River Jordan waters. Nevertheless, data given in Figure 6 indicate that an increase in annual means of the TP concentration in Lake Agmon waters accompanied the increase in crane populations during the period of 1993–2005 and 2007–2018, whilst from 2005 through 2011, lower levels of TP concentration were observed in Lake Agmon waters. A particular objective of this paper is to define the reason for an increase in the TP concentration in Lake Agmon waters. An attempt at defining the contribution of crane droppings as a major impact on the increase in the TP content in the Lake Agmon effluent is insufficient, and investigation of the seasonal fluctuation of the TP concentration in Lake Agmon waters denied the role of cranes as major contributors of phosphorus. Results given in Figures 7 and 8 clearly indicate that an increase in the TP concentration in Lake Agmon waters occurs in summer when cranes are not there. The seasonal pattern indicates the lowest concentrations of TP in winter months when cranes are there and increased concentrations in summer after migration of the cranes back to Europe for breeding (Figure 7). Figure 8 indicates that few cranes were observed in the Hula Valley [8] prior to 1993 when nutrient concentrations in the Jordan waters were high. The number of wintering cranes in the Hula Valley from 1993 and onward increased significantly (Figure 8; E. Naim and I. Rubin, NJF, personal communication), but nutrient concentrations in the Jordan waters declined (Figure 9). The uninterrupted continuity in the decline in nitrogen and phosphorus was prominent. Moreover, results presented in Figure 7 indicate the lowest TP concentrations in Lake Agmon waters during winter months, when cranes are present, whilst in the summer–

fall season, after the cranes migrate back to Europe and no cranes are observed in the Hula Valley, the TP concentration is elevated to its maximal level. Taking into account that during the period of 1970–1994, only single cranes were observed and, afterward, massive migration was recorded [8], the continuous trend of decline in nutrient concentrations (Figure 10) in the River Jordan was not interrupted or modified as a contribution that can be attributed to the impact of the crane population.

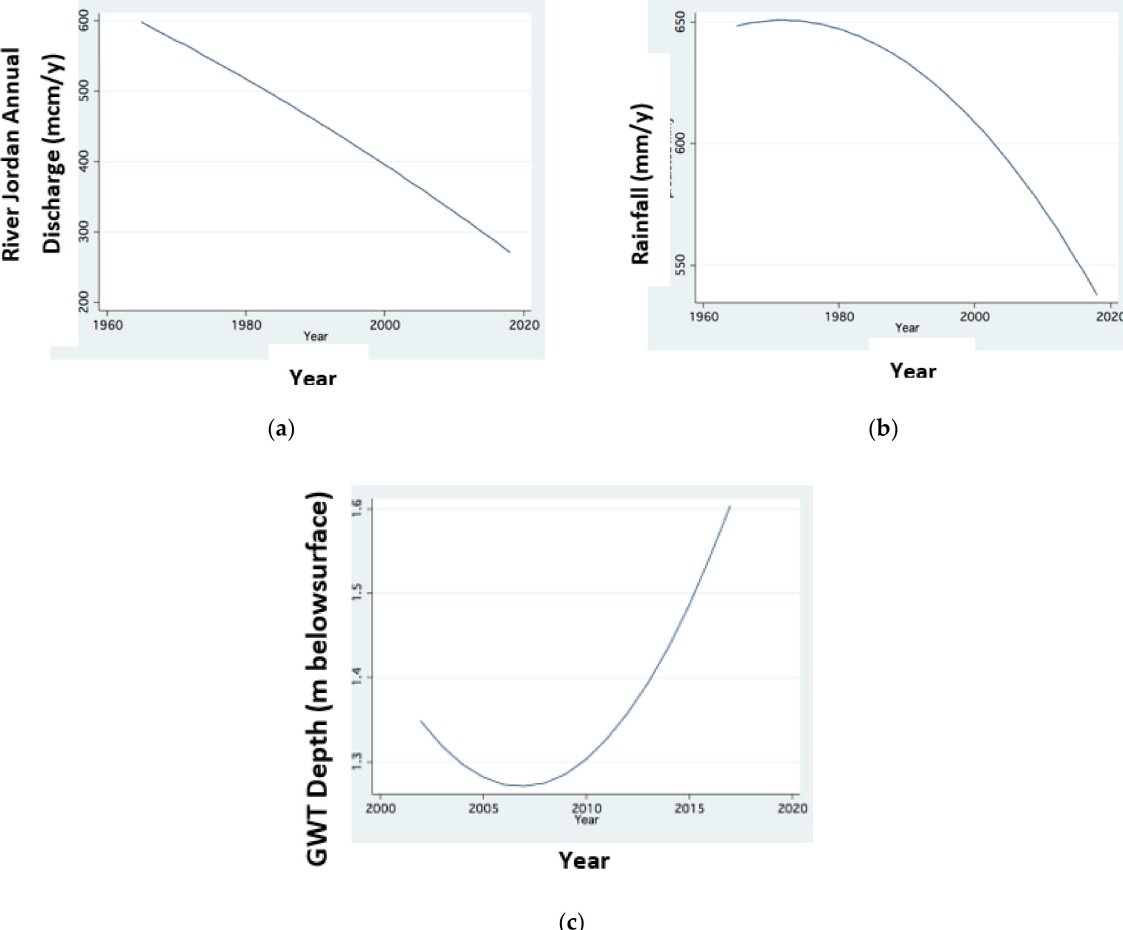

(a)

(b)

(c)

**Figure 2.** Climate change in the Lake Kinneret drainage basin: decline in the River Jordan annual discharge (106 m$^3$/year) during 1960–2020 (**a**), decline in annual precipitation (mm/year) (**b**), and lowering of the GWT depth (annual total Hula Valley average during 2003–2020) (**c**).

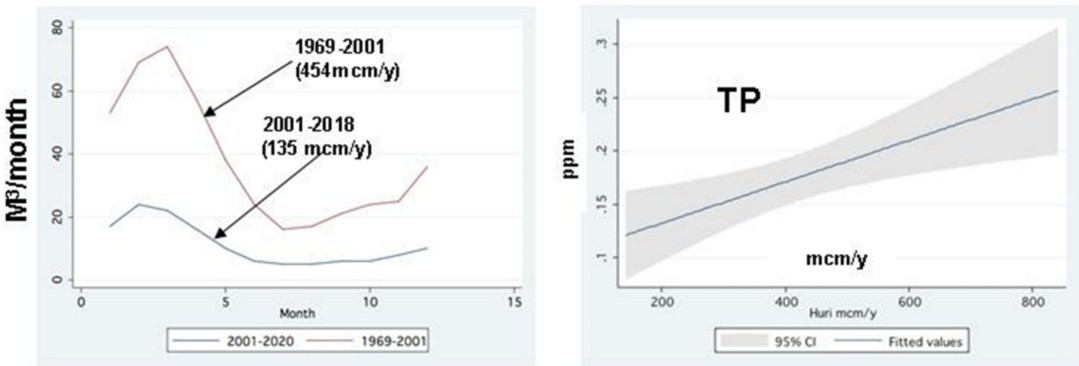

**Figure 3. Right**: linear regression (95% CI) between the River Jordan discharge (10$^6$ m$^3$/year) and the concentration (ppm) of total phosphorus (TP) in Jordan waters. **Left**: line scatter plot of the monthly discharge of River Jordan, averaged for two periods: 1969–2001 and 2001–2018.

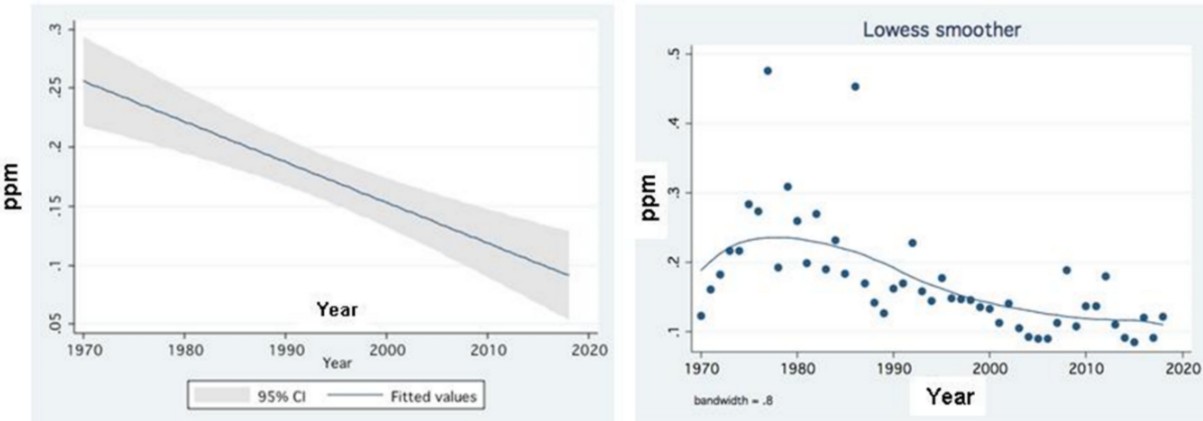

**Figure 4.** Temporal changes (1970–2018) in annual averages of total phosphorus (TP) concentrations (ppm) in Jordan waters: **Left**: linear regression (95% CI). **Right**: LOWESS smoother plot (bandwidth = 0.8).

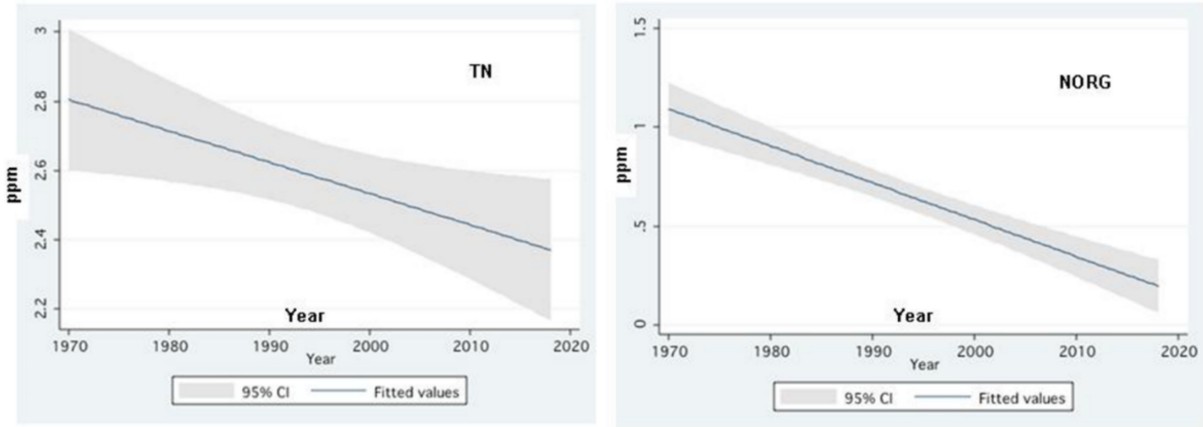

**Figure 5.** Temporal (1970–2018) changes in annual concentrations (ppm) of total nitrogen (**left**) and organic nitrogen (**right**) in Jordan waters, presented as linear regression (95% CI) between the concentrations and years.

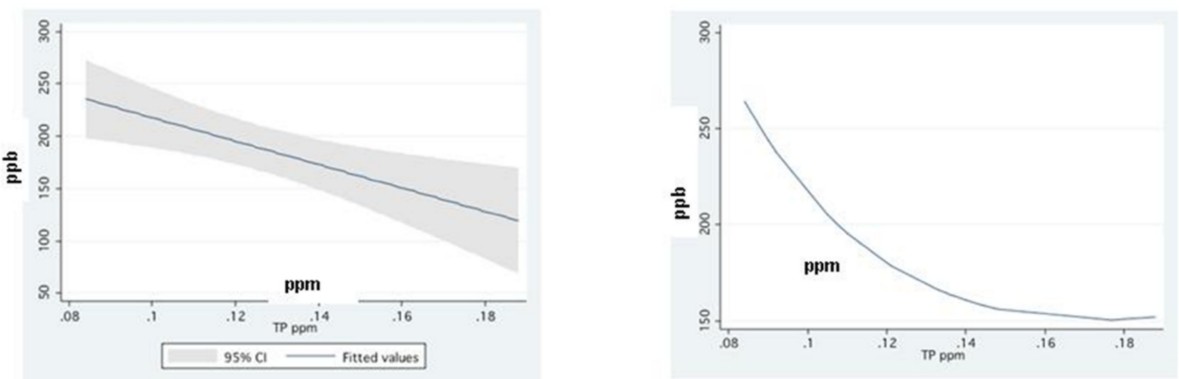

**Figure 6.** Relationship between annual (1993–2018) mean concentrations in the Lake Agmon outflow (ppb) and River Jordan waters (ppm). **Left**: linear regression (95% CI). **Right**: fractional polynomial regression.

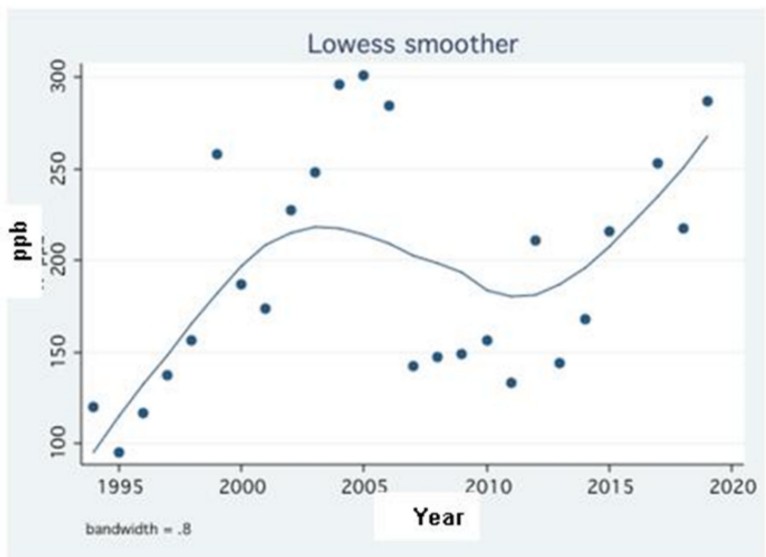

**Figure 7.** LOWESS smoother plot (bandwidth = 0.8) plot of the temporal (1994–2018) changes in annual averages of TP concentrations (ppb) in the Lake Agmon outflow.

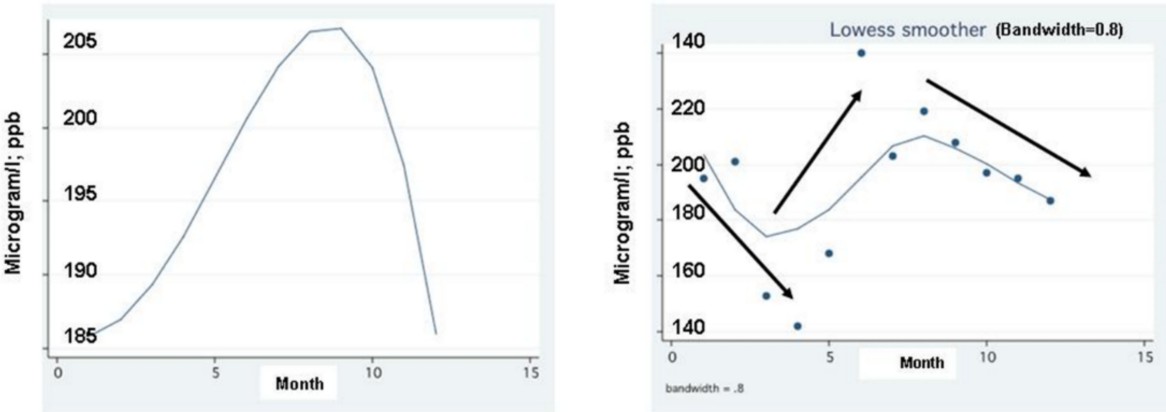

**Figure 8.** Monthly mean (1994–2018) fluctuations of TP concentrations (microgram/L; ppb) in the Lake Agmon outflow: **Left**: fractional polynomial regression. **Right**: LOWESS smoother plot (bandwidth = 0.8). Trends are arrowed.

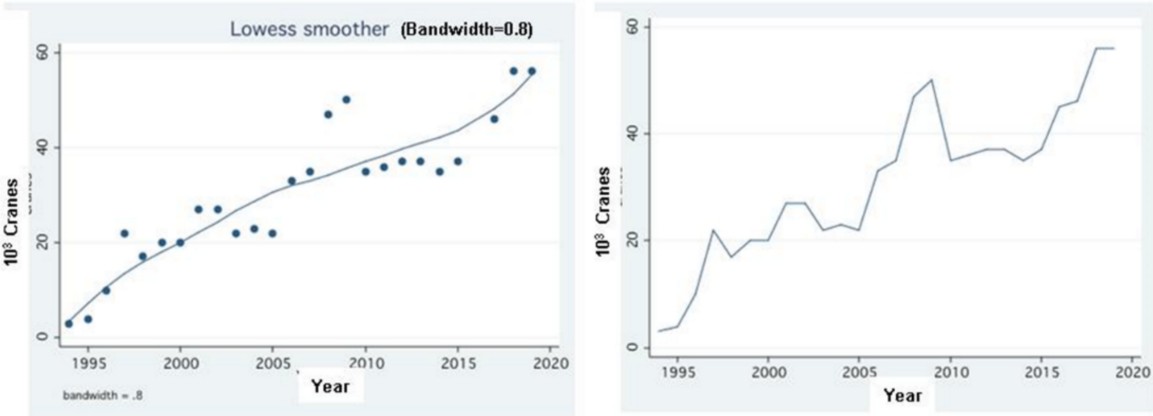

**Figure 9.** Temporal (1994–2018) changes in maximal counts of the wintering crane population in the Hula Valley. **Left**: LOWESS smoother plot (bandwidth = 0.8). **Right**: line scatter.

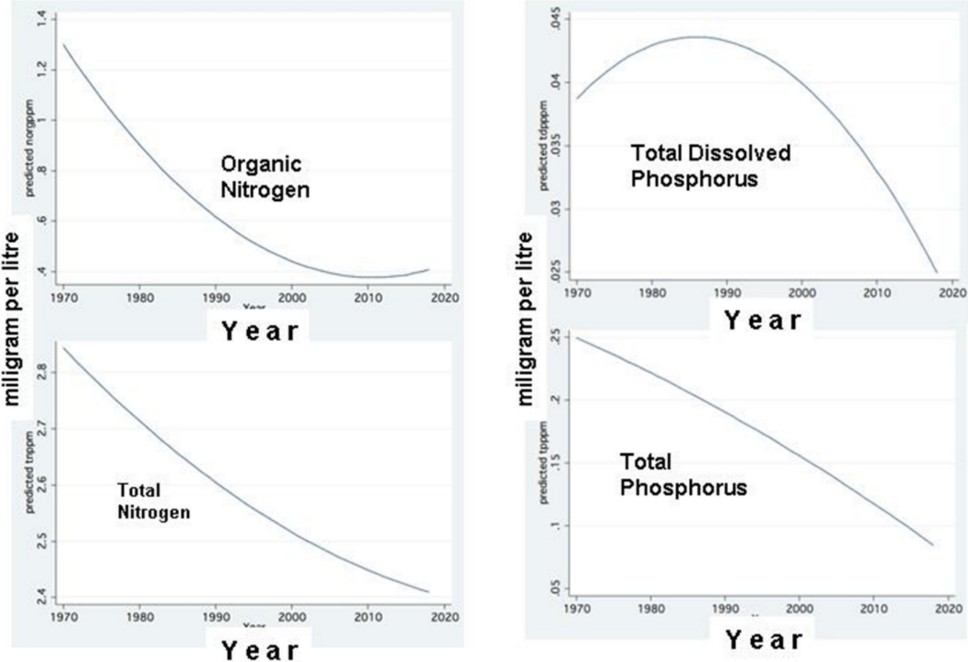

**Figure 10.** Temporal (1970–2018) changes in annual means of nutrient concentrations (mg/L; ppm) in River Jordan waters: **Upper left**: organic nitrogen. **Upper right**: total dissolved phosphorus. **Lower left**: total nitrogen. **Lower right**: total phosphorus. 1970–1993—no-crane period; 1994–2018—crane period.

The phosphorus mass balance in Lake Agmon as calculated for several years is presented next.

Input sources: Peat soil drainage area; major Canal Z = 101; Hula East and Reconstructed Jordan River route.

Effluent: Outflow from Lake Agmon into evacuation canals and direct pumping for irrigation.

Data on inflow and outflow of TN and TP are not available.

Due to technical difficulties, no sampling was done during 2016.

Two samples were collected on 18.9.2000 and 20.11.2000 when no cranes were present in the valley. The nutrient (TN, TP) concentrations were only affected by external inputs, and internal processes of geochemistry and vegetation decomposition were as follows (Table 2).

**Table 1.** Annual TP and TN mass load balances (tons/year) in Lake Agmon during 2008–2028: TN and TP inputs (inflow), output (outflow), and balance (output minus input) [14].

| Year | TN Inflow | TN Outflow | TN Balance | TP Inflow | TP Outflow | TP Balance |
|------|-----------|------------|------------|-----------|------------|------------|
| 2000 |           |            | 3          |           |            | 0.6        |
| 2004 | 248       | 19         | −230       | 1.03      | 0.430      | −0.6       |
| 2008 | 46        | 34         | −12        | 0.7       | 2.3        | 1.6        |
| 2009 | 78        | 47         | −31        | 1.1       | 2.5        | 1.4        |
| 2010 | 62        | 64         | 2          | 0.9       | 2.7        | 1.8        |
| 2011 | 112       | 55         | −57        | 0.9       | 2          | 1.2        |
| 2012 | 169       | 155        | −14        | 0.8       | 2.3        | 1.5        |
| 2013 | 150       | 113        | −37        | 0.7       | 1.8        | 1.1        |
| 2014 | 15        | 18         | 3          | 0.3       | 1.2        | 1          |
| 2015 | 75        | 43         | −33        | 1.2       | 2.2        | 1.1        |
| 2016 | 29        | 24         | −5         | 1.2       | 2.6        | 1.3        |
| 2017 | 121       | 30         | −91        | 1.1       | 1.8        | 0.7        |
| 2018 | 86        | 58         | −28        | 8.9       | 2.1        | 1.3        |

**Table 2.** TN and TP concentrations (ppm) in the Lake Agmon effluent measured on 18 September 2000 and 20 November 2000.

| Date | TN (ppm) | TP (ppm) |
|---|---|---|
| 18 September 2000 | 1.62 | 0.062 |
| 20 November 2000 | 2.16 | 0.471 |

Results in Table 2 indicate that from September, when submerged vegetation die-off (offset) initiates until November, when aquatic plants (sub- and above water) are intensively degraded, the TN and TP decomposed products enhance nutrient concentrations in the Lake Agmon outflow.

During September and November, cranes are absent in the Hula Valley.

Results given in Table 3 indicate a negative TPout mass balance during 2004. The average TP output for crane months (12, 1, 2, 3, 4) was 20 kg/month, whilst that for no-crane months (5–11) was 50 kg/month. During no-crane months, the Lake Agmon TP effluent load was higher than during crane months by 150%.

**Table 3.** Lake Agmon monthly TP mass (tons/month) balance (output minus input), TPin (tons/month), and TPout (tons/month) during 2004; yes = crane were present; no = cranes were absent.

| | TP Input | TP Output | TP Balance | Cranes |
|---|---|---|---|---|
| 1 | 0.24 | 0.06 | −0.18 | Yes |
| 2 | 0.07 | 0.01 | −0.06 | Yes |
| 3 | 0.02 | 0.001 | −0.02 | Yes |
| 4 | 0.03 | 0.02 | −0.01 | Yes |
| 5 | 0.09 | 0.06 | −0.03 | No |
| 6 | 0.12 | 0.01 | −0.11 | No |
| 7 | 0.08 | 0.02 | 0.06 | No |
| 8 | 0.09 | 0.05 | −0.04 | No |
| 9 | 0.11 | 0.07 | 0.04 | No |
| 10 | 0.08 | 0.07 | 0.01 | No |
| 11 | 0.1 | 0.07 | −0.03 | No |
| 12 | 0.01 | 0.01 | 0 | No |
| Total Annual | 1.03 | 0.43 | −0.06 | Yes |

Results in Table 4 indicate higher TPout concentrations (given in ppb in Table 4) in no-crane months (5–11) than those during crane months (12, 1, 2, 3, 4). The monthly average for crane months was 228 ppb, while that for no-crane months was higher—292 ppb. An average periodical (1994–2018) increase in the TPout concentration by 28% in no-crane months than in crane months was recorded.

**Table 4.** Monthly averages (1994–2018) of TPout concentration (ppb) in the outflow from Lake Agmon.

| Month | TP (ppb) | Cranes |
|---|---|---|
| 1 | 222 | Yes |
| 2 | 258 | Yes |
| 3 | 232 | Yes |
| 4 | 183 | Yes |
| 5 | 222 | No |
| 6 | 222 | No |
| 7 | 204 | No |
| 8 | 317 | No |
| 9 | 473 | No |
| 10 | 279 | No |
| 11 | 325 | No |
| 12 | 245 | Yes |
| Annual Average | | |

The TP balance in Lake Agmon was evaluated as follows: The TP concentration and water discharge were routinely (weekly) monitored in the three major Lake Agmon water inflow sources: Canal Z, Hula East, and Reconstructed Jordan (see maps in Figures 1 and 2). The loads were computed as multiplication of the TP concentration by the water discharge. An example of year 2003's TP balance is given in Tables 5–7.

**Table 5.** Monthly averages of TP concentrations (ppb) in the water sources of Lake Agmon: Canal Z, Hula East, and Reconstructed Jordan (see Figures 1 and 2) during 2003.

| Month | Canal Z | Hula East | Reconstructed Jordan | Lake Agmon Effluent (TPout) |
|---|---|---|---|---|
| 1 | 56 | 0 | 6 | 54 |
| 2 | 28 | 12 | 11 | 44 |
| 3 | 0 | 6 | 0 | 70 |
| 4 | 26 | 9 | 119 | 22 |
| 5 | 33 | 0.3 | 89 | 113 |
| 6 | 57 | 13 | 40 | 31 |
| 7 | 24 | 5 | 48 | 28 |
| 8 | 91 | 8 | 71 | 98 |
| 9 | 15 | 3 | 12 | 7 |
| 10 | 15 | 4 | 31 | 19 |
| 11 | 35 | 17 | 8 | 63 |
| 12 | 3 | 19 | 10 | 24.2 |

**Table 6.** Monthly discharges ($10^3$ m$^3$) of water inflows through the sources into Lake Agmon: Canal Z, Hula East, and Reconstructed Jordan during 2003.

| Month | Canal Z | Hula East | Reconstructed Jordan | Outflow |
|---|---|---|---|---|
| 1 | 500 | 0 | 70 | 600 |
| 2 | 400 | 100 | 150 | 300 |
| 3 | 0 | 30 | 0 | 500 |
| 4 | 300 | 60 | 420 | 400 |
| 5 | 400 | 2 | 410 | 700 |
| 6 | 500 | 120 | 380 | 300 |
| 7 | 600 | 60 | 410 | 300 |
| 8 | 800 | 60 | 750 | 500 |
| 9 | 700 | 100 | 680 | 200 |
| 10 | 400 | 100 | 600 | 300 |
| 11 | 400 | 80 | 110 | 500 |
| 12 | 40 | 80 | 200 | 1000 |
| Total Annual | 5040 | 792 | 4180 | 5600 |

**Table 7.** Monthly inputs of TP loads (kg/month) into Lake Agmon through major sources of water inflows into Lake Agmon: Canal Z, Hula East, and Reconstructed Jordan during 2003.

| Month | Canal Z | Hula East | Reconstructed Jordan | Total Input | Total Output | TP Balance |
|---|---|---|---|---|---|---|
| 1 | 28 | 0 | 0.4 | 28.4 | 32.4 | 4 |
| 2 | 11.2 | 1.2 | 1.7 | 13.1 | 13.2 | −0.9 |
| 3 | 0 | 0.2 | 0 | 0.2 | 35 | 34.8 |
| 4 | 7.8 | 0.5 | 50 | 59.3 | 8.8 | −49.5 |
| 5 | 13.2 | 0 | 36.5 | 49.7 | 79.1 | 29.4 |
| 6 | 28.5 | 1.6 | 15.2 | 45.3 | 9.3 | −36 |
| 7 | 14.4 | 0.3 | 19.7 | 34.4 | 8.4 | −26 |
| 8 | 72.8 | 0.5 | 53.3 | 126.6 | 49 | −77.6 |
| 9 | 10.5 | 0.3 | 8.2 | 19 | 1.4 | −17.6 |
| 10 | 6 | 0.4 | 18.6 | 25 | 5.7 | −19.3 |
| 11 | 14 | 1.4 | 0.9 | 16.3 | 31.5 | 15.2 |
| 12 | 1.2 | 1.5 | 2 | 4.7 | 24.2 | 19.5 |
| Total Annual | 207.6 | 7.9 | 206.5 | 422 | 298 | −124 |

Results given in Table 7 indicate a much higher (TP balance of 132.5 kg) internal TP input during the no-crane months (5–11), whilst during the crane months (12, 1, 2, 3, 4), the total balance was positively 50.4 kg. Consequently, during summer months after crane migration, there was an internal source of TP, most likely aquatic vegetation. The role of decomposed plant matter in the Lake Agmon outflows during spring–summer–fall months has been published in earlier studies [24–27]. A distinct elevation in drifted/flushed phosphorus within the Hula Valley was documented (Figure 6, Table 8) [13–18].

**Table 8.** Monthly means of TPout concentrations (ppb), as calculated for two periods, 1994–2005 and 2006–2020, when the migrated crane population maximum size varied between 17,000–27,000 and 33,000–56,100, respectively.

| Month | 1994–2005 | 2006–2020 |
|:-----:|:---------:|:---------:|
| 1 | 165 | 251 |
| 2 | 189 | 305 |
| 3 | 155 | 272 |
| 4 | 117 | 215 |
| 5 | 122 | 284 |
| 6 | 133 | 269 |
| 7 | 156 | 233 |
| 8 | 206 | 408 |
| 9 | 177 | 575 |
| 10 | 185 | 332 |
| 11 | 240 | 381 |
| 12 | 192 | 287 |

Information given in Table 8 indicates a higher concentration of TPout during 2006–2020 than earlier (1994–2005). Moreover, Table 4 indicates an average TPout concentration of 228 ppb during the crane months (12, 1, 2, 3, 4) and 349 ppb during summer–fall months of August through November. That is a percentage elevation of 53% in a period when cranes are absent. It is therefore suggested that processes initiating these changes are not related to the presence of cranes in the valley. Statistical evaluation of the correlation between the population size of migrating cranes and the TP balance in Lake Agmon was found to be insignificant. In contrast, a simple regression ($r^2$ and *p*-value) test between TPin, TPout, and TPbalance indicated significant relationships: (1) TPout vs. TPin, (2) TPbalance vs. TPin, and (3) TPbalance vs. TPout; values of $r^2$ were above 0.6507 and $p < 0.0001$. It is therefore likely that phosphorus dynamic processes in Lake Agmon, including sources and effluents, have terrestrial traits and are mostly affected by erosion, geochemistry, and vegetation but are not significantly affected by cranes.

A full-year cycle (14 months) of monthly sampling of groundwaters followed by chemical analysis of nutrient contents was carried out. Results of underground water phosphorus level (GWT) in the subterranean waters indicated a hydrological gradient directed north to south (northern higher level). Nevertheless, the phosphorus content indicates a reversible gradient: higher level of southern subterranean phosphorus—in other words, accumulation of phosphorus in the southern underground capacity. Despite cumulative migration of phosphorus from north to south, its stock is limited, probably resulted by deeper continuity of the migration. This assumption has not yet been approved, but the contribution of phosphorus through underground routes into the Jordan River and further into Lake Kinneret was not indicated. The flux of phosphorus from the subterranean sources into Lake Agmon through advective inflows was documented as negligible [28,29].

## 4. Discussion

There is a difference between nitrogen and phosphorus dynamics in Lake Agmon. The lake is a nitrogen sink. Nevertheless, previous studies have confirmed partial removal of nitrogen from the Lake Agmon waters through denitrification and sedimentation. The continuity of a positive TPbalance indicates supplemental TP resources other than peat soil drainage, possibly crane droppings and re-suspension and/or submerged macrophytes.

Local daily migration of cranes in the Hula Valley indicates terrestrial allocation during the daytime and in the Lake Agmon shallows at night while excretion of their droppings into the water. A quantitative monitoring of the seasonality of above and sub-surface aquatic vegetation was carried out during 1997–2004 [27]. The submerged vegetation onset started in April and peaked through July–August, followed by offset with dieback disappearance in December. During winter time (December through April), the aquatic plant biomass is negligible. Results indicate an average dry weight biomass of 456.4 tons (min.-max. −140–817) contained 0.9 tons of TP (max.–min. −0.3–1.2) and 7.4 tons of TN (max.-min. −2.7–10.5). The supplemental TP and TN loads to Lake Agmon through aquatic vegetation as related to soil drainage input were 10% and 8%, respectively.

Lake Agmon was constructed during 1993 and filled with water in the summer of 1994. The ecological stability during the first years was partly flexible, and the Lake Agmon ecosystem was not yet stable. Therefore, nutrient mass balances were changed widely. For example, during the early 2000s, when the crane population was below 20,000, the TN and TP mass balance differed from the later period of 2008–2018. The TP balance prior to the crane migration was positive, and four years later, it was negative; therefore a significant contribution of TP by cranes is not suggested. Long-term records (1995–2018) indicate average (high range of SD) TN and TP concentrations in the Lake Agmon outflow of 4.67 and 0.17 ppm, respectively. Although the total mass removal of nutrients through the Lake Agmon system is not high, how much is affected by crane migration was not primarily predicted. Nevertheless, the seasonality of vegetation growth and consequent nutrient loads was also unpredictible. The ecological significance of a newly constructed system, such as the Hula Reclamation Project, accompanied by crane migration initiated the present study.

The winter migratory cranes, which are fed supplementally by tons of corn seeds, are an essential source of phosphorus for the Lake Kinneret ecosystem. The four-month winter visit of cranes might therefore lead to lake pollution by phosphorus. The dynamics of water-mediated phosphorus input into Lake Kinneret is driven by the hydrological runoff and subterranean linkage between the Hula Valley and Lake Kinneret. Until the early 1990s, cranes were almost absent in the Hula Valley, excluding a few individuals. Since then, the valley has been populated annually from November through March by increasing numbers of cranes, up to 50,000 in the winter of 2019–2020 [8]. Leftover harvested peanuts attract the cranes in October–November. Peanut crops have been found to yield significantly high revenues and are suitable to be grown on the heavy-organic peat soil in the Hula Valley. In early to mid-December, the cranes start looking for other sources of food, thereby causing damage to winter crops. An efficient method aimed at prevention of agricultural damage has been improvised: Money is allocated as rental for a 40 ha field block in the valley, which serves as a feeding station for the cranes; here, corn seeds are purchased and fed to the cranes twice a day. Feeding starts in late December and continues until early March, when the cranes fly back to Europe for breeding [8].

This has proved to be efficient, but it has a costly disadvantage, since there is the tendency of phosphorus leakage into Lake Kinneret. This paper aims at suggesting ways of removal of the pollution parameter but also leaves room for further consideration of the unresolved issue of the financial cover source for the corn seed purchase.

A brief search of the literature about phosphorus metabolism of birds indicates a wide range of its excretion [30,31] Although cranes are not typical water fowls due to their day–night stay discrimination between terrestrial and aquatic habitats, they are not the only Lake Agmon inhabitants. Among other inhabitants are ducks, herons, pelicans, cormorants, mallards, and seagulls [32]. Big flocks of migratory birds, such as cormorants, pelicans, seagulls, and mallards, are distributed among northern water bodies (fishponds, reservoirs, Lake Kinneret, temporal ponds, regional rivers), whilst the cranes create huge flocks located in the Hula Valley and presently (>2004) assembled in Lake Agmon during nighttime. Pelicans create big flocks but stay for a short period (2–3 weeks) partly on terrestrial land and partly in Lake Agmon. Cormorants are scattered in singles and locally migrate to

Lake Kinneret. Conclusively, cranes are the major group to be defined as potential TP contributors to Lake Agmon input loads.

The following data about phosphorus excretion by fowls were considered: *Mallard platyrinchos* and *Larus ridibundum*, 1–0.1 gP/bird/day [30]; cultured poultry broilers, 0.2–0.3 gP/bird/day [31]; and different migratory water fowls, 5.24 gP/bird/day (cranes and pelicans) and 3.5 gP/bird/day (cormorants) [33–37]. For the maximal effective value, the P excretion of cranes at 5.24 gP/bird/day was considered. Considering the presence of $50 \times 10^3$ cranes for 150 days (November through March) with 100% influx of their droppings into Lake Agmon waters, the annual P load from the birds would be (5.24 g) × (50,000 birds) × (150 days), which is equivalent to 39.3 tons. This is doubtless a significant extra load, which is almost 45% of the total measured annual input into Lake Kinneret from the River Jordan discharge. Nevertheless, this additional P load is not confirmed to be an extra loading to Lake Kinneret and likely also not a source of P to Lake Agmon. However, it was confirmed that this excreted P accumulates in the peat soil within the Hula Valley [15,38–42]. It is likely that P accumulation in the uppermost peat soil layer is probably not unlimited, and P migrates into the shallow subterranean groundwater table and most probably even much deeper. The role of migratory water fowls as nutrient vectors in managed wetlands was documented by Post et al. [43]. However, cranes as P vectors of corn seed P mediated into Lake Agmon and further on into Lake Kinneret were not confirmed.

The conflict between cranes and agriculture is well known and highly documented [8]. Cranes are protected by international laws; shooting them is illegal, and thus deportation should be done without shooting. A collaborative solution between farmers, nature authorities, water managers, landowners, and regional municipalities was contracted and implemented. Cranes that land prior to mid-December are deported without shooting, with the aim to reduce the number of potential feeders, thereby preventing damage and reducing the cost of corn seeds. Corn seed feeding starts in mid-December. Cranes are not nocturnal, and their diurnal behavior includes feeding during the day; then they spend the full-darkness period (about 12–14 h) in Lake Agmon. The daily schedule of the cranes comprises 10–12 terrestrial day hours and a 12–14-h night period in Lake Agmon. Their noisy and remarkable local migration is aimed at reducing their vulnerability to predators. It is likely that most of the crane excretion takes place in the Lake Agmon waters at night. Studies carried out on migratory birds documented an increase in daily foraging activity with daytime (light time) prolongation, and the opposite is reasonable because daytime activity levels depend on light conditions (sun elevation) [44]. Moreover, Haynes et al. [45] documented intensive night activity of foraging and copulation by many species of water (wetland) birds, such as gulls (Laridae). Kostecke et al. [46] documented nocturnal time activity budgets consisted of foraging (62%) and resting (20%). A greater percentage of time was devoted to foraging and aggression, and less was spent resting at night [46]. Therefore, searching for the TP migration rate through Lake Agmon effluents is justified. Moreover, since 1994, the routine sampling of the Lake Agmon outflow has been carried out on a weekly basis in the mornings. The relevance of these samples to the evaluation of the impact of crane excretion on Lake Kinneret inputs might be criticized because the nocturnal stay of the birds is terminated before routine sampling. To avoid a biased conclusion, a calculation of the residence time (RT = inflow/volume) in Lake Agmon was done: the mean (2008–2018) value of the Lake Agmon daily inflow was $25.2 \times 10^3$ m$^3$, the Lake Agmon surface area is 82 ha, and the mean depth is 20 cm. Consequently, the Lake Agmon total volume is $164 \times 10^3$ m$^3$ and RT = 6.5 days, confirming the minority of daily flushing out of excreted crane droppings with accumulative potential. A major part of TP content in the crane droppings is due to the daily ingestion of food (corn seeds) collected outside Lake Agmon and results from resuspension carried out by bird stepping. Mass balances for TP in Lake Agmon indicate a negative regime, i.e., outflow is higher than inflow loads. These results, together with the results presented in Figure 7, support the conclusion about other sources besides crane droppings of the summer increase in TP in the Lake Agmon

effluent. Moreover, during the period of 1970–1993, before the crane era (CE), the mean TP concentration was 0.224 ppm, whilst later (1994–2018), during the CE, it was 0.126 ppm. Conclusively, an increase in the TP concentration in Lake Agmon due to the activities of cranes is not confirmed. The Hula Valley annual contribution of TP and TN through the hydrological system of Lake Agmon, after the implementation of the Hula Reclamation Project (1996–2008), confirms the following: 1.2 and 26 tons of TP and TN, respectively, were removed from the Lake Kinneret external loads, which comprised 1.7% and 2.4% of the total loads of TP and TN, respectively [3,7,12,25,26]. Long- and short-term geochemical research on phosphorus and nitrogen dynamics within the peat soil in the Hula Valley, as well as in drainage canals in the vicinity of Lake Agmon [15,38–42] and in the subterranean waters, indicated high TP and TN levels. Nevertheless, the composition and dynamics of these pollutants in the River Jordan waters was not positively related to the soil and underground flows from the geochemical findings.

This achievement initiated benefits for both landowners and farmers by generating income from about half a million bird watchers (charged visit), whilst the Hula Valley effluents were not significantly deteriorated. Moreover, the cranes were allotted a night habitat underneath the terrestrial eucalyptus trees, where they became vulnerable to predators (fox, wolf, mongoose, jackal). Therefore, the bird flocks changed from the night habitat to the protected refuge site in the newly created shallow Lake Agmon–Hula.

The crane management project (deportation and feeding) was designed to be part of a comprehensive objective aimed at enhancing ecosystem sustainability. The solution is conclusively summarized as follows: to reduce the agricultural damage on a fixed land site where cranes have gathered during the day to feed on corn seeds, leaving this area for the shallow lake at night, where they would be protected from predators. Bird watchers visits and the Hula Project lead to nutrient removal from Lake Kinneret loads. This crane management project represents an efficient "marriage" by establishing a balance between bird and limnological interests for the prevention of eutrophication in Lake Kinneret.

The Hula Reclamation Project was aimed at ensuring sustainability of modified ecosystems by formulation of a conflict between agricultural development, Lake Kinneret water quality protection, and nature conservation. The tension between farmers, water managers, and nature preservers was reduced, and there was collaboration instead. The outcome of the HRP was renewal of the ecosystem, which had become a tourist attraction, enriching the biological diversity with approximately 300 species of birds, including 40,000–50,000 wintering cranes annually, 40 species of water plants, and 12 species of fish. The new ecosystem of the shallow Lake Agmon–Hula with surrounded safari habitats ecosystem became a tourist attraction and maintenance investment. The potential contributors of water-mediated phosphorus include the following: Lake Kinneret headwaters, Lake Agmon–Hula, crane droppings, aquatic vegetation, and the major peat-soil-drained water pathways in the Hula Valley.

A tentative semi-conclusion at this point indicates two major points: Phosphorus removal from the Lake Kinneret loads through the Lake Agmon system is minor, and phosphorus enrichment occurs in the late summer–fall season. A working hypothetical assumption suggests sources of phosphorus other than crane droppings and resuspension. These other sources are peat soil geochemical flushing introduced into the Lake Agmon water column through water inflow. The second source is phosphorus uptake from the sediments, incorporated into plant tissues and later, as a result of plant decomposition, into the Lake Agmon water column. Earlier studies [47–49] and long-term records have indicated peat-soil-bounded phosphorus release mostly during summer dryness, which enhances linkage breaks between organic phosphorus and peat soil particles [50,51]. The dynamics of submerged high plants and algal organisms was documented. The most abundant organisms were, among others, submerged *Potamogeton* spp., *Najas* spp., *Myriophillum* sp., filamentous mats and scum of Cyanobacteria and Chlorophyta plants that stand out of the water, *Typha* sp., *Phragmites* spp., and others [52]. The seasonal dynamics of their growth rate, dry weight, and nutrient (N, P, K) content were documented. These measurements

clearly indicated onset of vegetation growth during late March–early April and offset during September–November. It was confirmed that the plant degradation during fall contributes a significant amount of phosphorus as dissolved and particulate forms [52]. Earlier studies [18] have confirmed a significant elevation in the phosphorus concentration in the Lake Agmon waters during summer months. Moreover, the Hula Reclamation Project annual reports (1994–2018) documented a continuous increase in phosphorus concentration during summer months in the major canal that drains organic peat soil. The TP elevation is due to vegetation degradation and not crane activity. Therefore, to achieve a beneficial ecological project of crane attraction, the potential damage due to phosphorus pollution is eliminated and prevention of agricultural damage remains a major target. The long-term record confirmed that such an objective is feasible but expensive. Goodwill and friendly relations between all partners involved—landowners, farmers, hydrologists, and nature and Lake Kinneret protectors—make this achievement feasible.

The combination of beneficial crane attraction as bird watcher visits and hydrological management was confirmed. The merit of cranes as a complementary achievement of eco-tourism policy to the hydrological management of the Hula Valley was also confirmed. Data shown in Figure 11 indicate similar enhancement of the TP concentration in the Lake Agmon effluent during 1994–2018 (Figure 13, left panel) without a particular impact of the crane migration. Moreover, a close relationship is shown (Figure 12) of the temporal pattern of changes between TP input and TP balance in Lake Agmon. No significant crane impact on these parameters is therefore suggested. The TP sources for the input loads are terrestrial, with probably a minor impact by cranes. Monthly means of the TP concentration in the Lake Agmon effluent (Figure 13, Figure 14 right panel) justify the conclusion of significant TP concentration enhancement resulting from summer decomposition of aquatic macrophytes when cranes are absent. Moreover, during 1994–2018, the TP concentration during winter months (12, 1, 2, 3, 4) was quite stable (240 ppb), declined in May–June, followed by an abrupt and sharp increase (250–360 ppb) in summer–fall (July–November). Conclusively, the domination of TP concentration in the Lake Agmon waters is due to input (winter) and macrophyte decomposition (summer–fall) (Figures 14 and 15). Although the possibility of phosphorus pollution of the Lake Agmon waters by cranes was rejected, their agricultural damage is significant, and the solution to this damage is costly. Crane migration starts in late October, and they stay until late February to early March, for a total of 150 days. As of mid-December, cranes feed on the leftover peanut harvest and deportation is implemented by the landowners. From mid-December until backward natural migration, they are fed corn seeds in designated blocks, accompanied by partial low-level deportation from other blocks. During the 2003/4–2005/6 season, 250 tons of corn seeds were fed to the cranes in their designated blocks, whilst in 2020/21, 361 tons were fed to them. The crane population in the Hula Valley increased from below 20,000 before the 2000s to 50,000 in 2021 (Figure 8). Moreover, the number of visitors and bird watchers increased dramatically from <200,000 in the years 2005–2008 to 500,000 in 2021 (E. Naim and I. Inbar, JNF-KKL unpublished data).

The significance of the controversial conflict between the three ecological compartments is better understandable by supplemental information about the financial management of the crane project during the winter of 2020–2021 (O. Barnea, Director of the Agricultural Society of the Upper Galilee municipality, unpublished):

Total fed (107 days) corn seeds—361 tons (cost 225,000 US\$)

Total cost of crane deportation (6552.3 ha)—$2 \times 10^6$ US\$.

Conclusion: Crane migration is an attractive issue incorporated within the ecological concept of the land-use policy integrating eco-tourism and hydrological management, but it is costly.

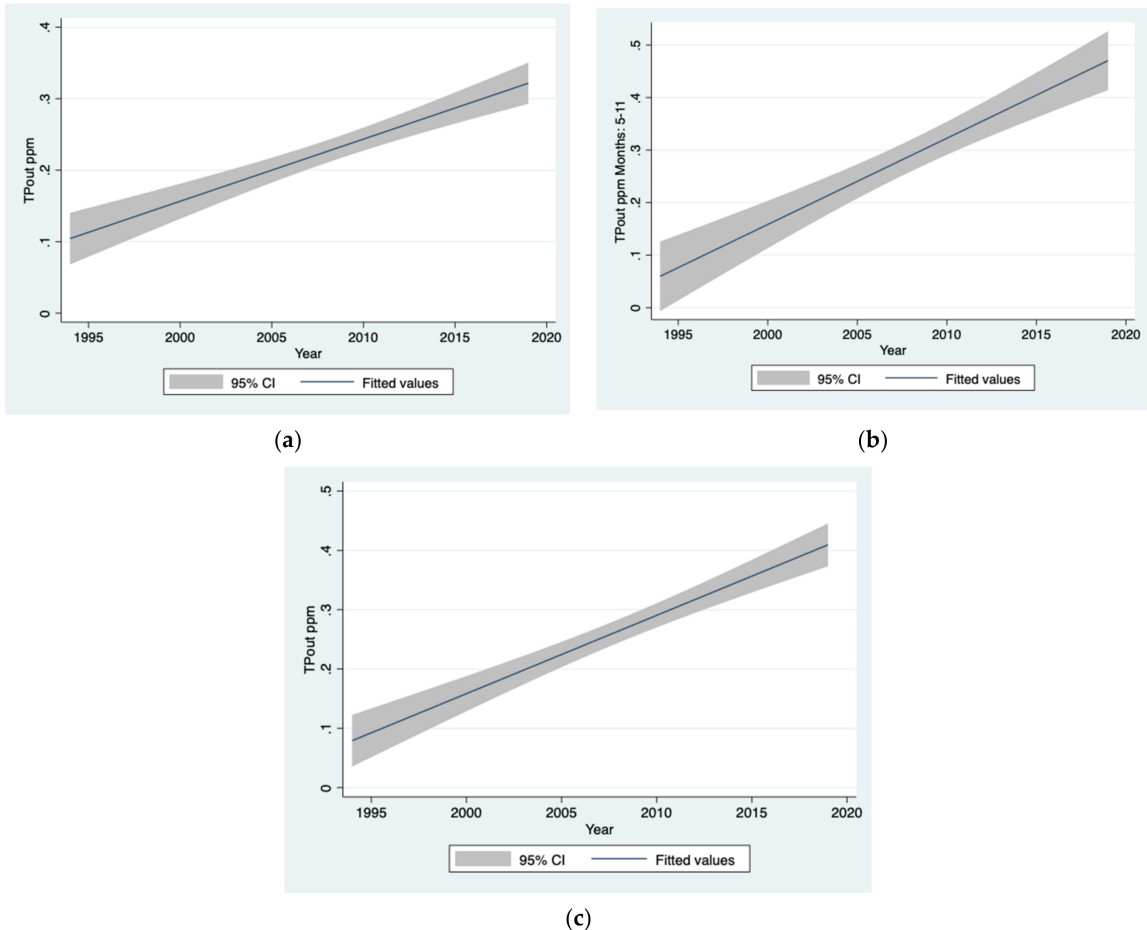

**Figure 11.** Linear regression (95% CI) between monthly means of TPout concentration (ppm) during crane months (12, 1, 2, 3, 4) (**a**) and no-crane months (5–11) (**b**) in 1994–2018. Linear regression (95% CI) between annual means of TPout concentrations (ppm) in (1994–2018) (**c**).

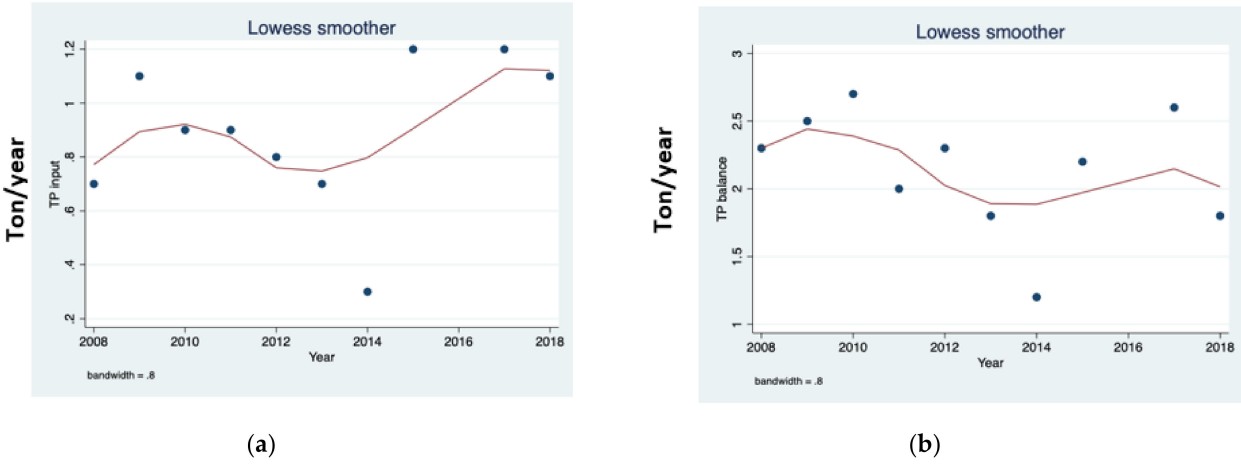

**Figure 12.** LOWESS smoother plot of trend in changes in annual loads (tons/year) of TPin (**a**) and TPbalance (**b**) during 2008–2018.

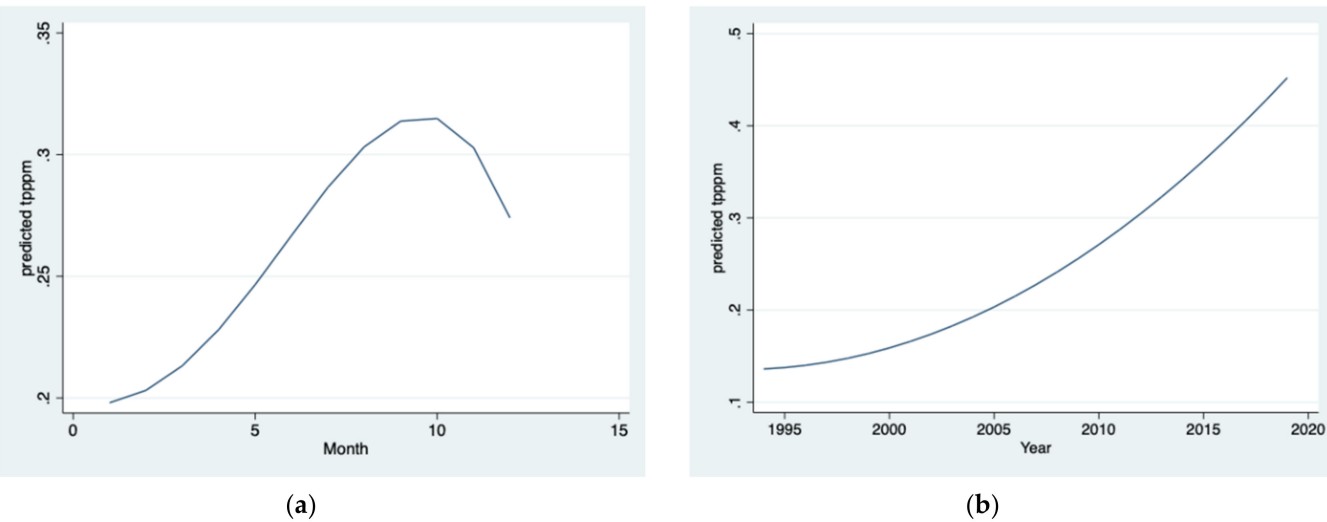

**Figure 13.** Fractional polynomial regression between monthly (**a**) and annual (**b**) means (1994–2018) of TPout concentrations (ppm).

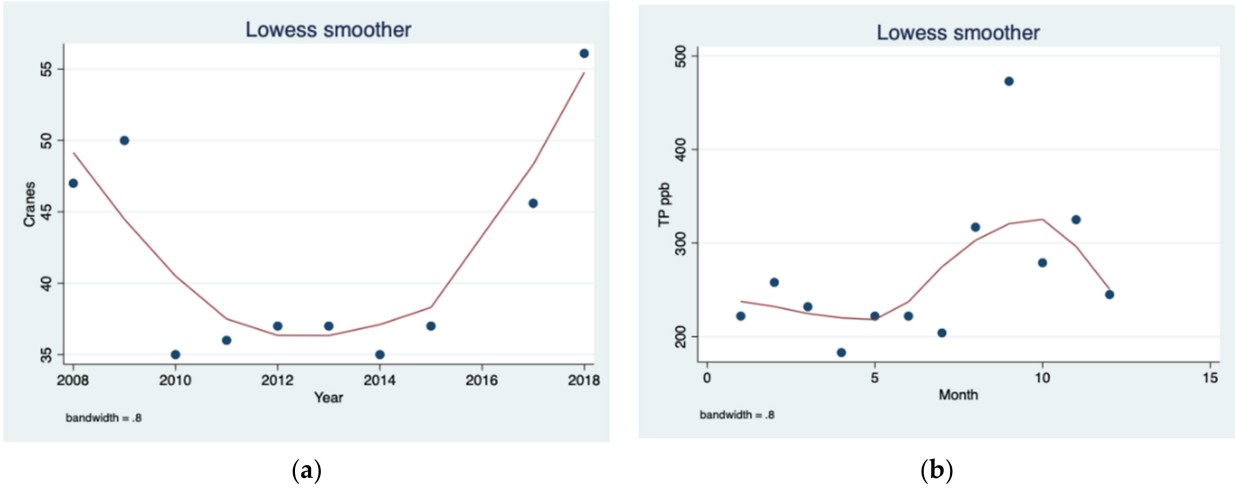

**Figure 14.** (**a**) Maximum cranes counted ($10^3$). (**b**) LOWESS smoother trend plot of changes in monthly means of TPout (ppb) during 1994–2018.

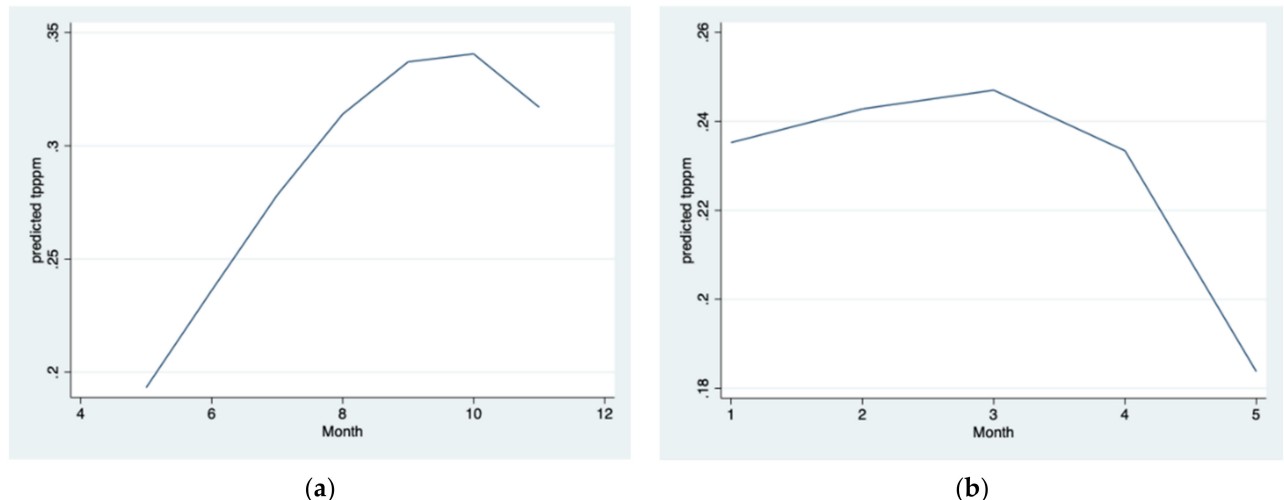

**Figure 15.** Fractional polynomial regression between monthly means (1994–2018) of TPout concentration (ppm) and no-crane months (5–11) (**a**) and crane months (12, 1, 2, 3, 4) (**b**).

## 5. Conclusive Remarks

Documented information on the contribution of TP from the Hula Valley due to winter migration of cranes does not include water quality damage in Lakes Agmon and Kinneret. Geochemical research on the phosphorus dynamics in the organic peat soil of the Hula Valley indicated a high content in the upper layers of the crane-feeding land plot as well as in adjacent drainage canals [15,38–42]. Nevertheless, these results were not long term (multi-annual). Short-term TP elevation in the Lake Agmon waters were documented, but the TP concentration diminished significantly for the rest of the time. It is therefore unlikely that the crane presence poses long-term risks to water quality in both Lakes Agmon and Kinneret. Moreover, data shown in this paper confirm the multi-annual elevation in the TP concentration in Lake Agmon, but analysis of TP seasonal fluctuation confirms that this increased concentration could be attributed to aquatic plants and not cranes. Underground-water-mediated phosphorus migration from northern to the southern Hula Valley was indicated earlier but not later. However, regarding the fate of accumulated phosphorus in the shallow depths of the southern Hula Valley, an unconfirmed hypothesis was defined: this stock phosphorus migrates into much deeper strata. A study was carried out in the early 1970s as part of the Lignite Project; deep (below 20 m) lignite waters were analyzed, and difficult breakable organic compounds containing phosphorus were found. Another unconfirmed assumption was raised later about the possibility of deep-layer vertical migration toward the south. Conclusively, the underground-water-mediated and, probably, part of the runoff-mediated phosphorus in the drainage canals in the Hula Valley is not fluxed into Lake Kinneret.

**Funding:** This research received no external funding.

**Acknowledgments:** The construction and routine operation of the Hula Reclamation Project and the agricultural management in the Hula Valley were implemented and routinely monitored by the Keren Kayemet LeIsrael, Jewish National Foundation; MIGAL, Scientific Research Institute; the Upper Galilee Municipality; the Agriculture Society, Upper Galilee; and the Israeli National Water Authority. Data, technical assistance, and financial support were implemented by KKL-JNF, MIGAL, and the Israeli National Water Authority. Warm personal thanks are due to E. Naim, I. Inbar, V. Orlov-Levin, M. Meron, Y. Tsipris, E. Yasur. T. Natanzon, I. Barnea, and G. Eshel. The crane project was (27 April 2021) presented at the Galilee Research Conference, Tel Hai College, by Y. Leshem, D. Alon, O. Barnea, and M. Gophen.

**Conflicts of Interest:** The author declare no conflict of interest.

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
