# Peer review of "Hydrology and Cranes (Grus grus) Attraction Partnership in the Management of the Hula Valley—Lake Kinneret Landscape"

_hydrology, doi:10.3390/hydrology8030114_

Round 1

Reviewer 1 Report

The paper has been substantially improved and may be published. I found a few small things to improve:

lines 108-110: Kap[lan and improper spaces

Figure 1 standardize the font in the inscription YEAR

use a single margin throughout the text

put new tables in order:
- use a uniform font
- spatially separate individual tables (now everything merges into one block)
- if possible, do not divide a single table between pages

line 702 issue 4 not $

literature 44-47 in different style

Author Response

Comments and corrections given by reviewer #1  are  agreed by the author and  incorporation was implemented/

Reviewer 2 Report

Although there is coherence and a good use of the english language in this paper, the structure appears to be quite stuffed, there is almost no proper use of paper structure, separation of paragraphs or organization of tables.

This need to be corrected to be able to understand properly the paper.

Author Response

Comments and required corrections are agreed by the author and implemented

in red color in the revised version

This manuscript is a resubmission of an earlier submission. The following is a list of the peer review reports and author responses from that submission.

Round 1

Reviewer 1 Report

The article is interesting and includes an analysis of the effects of a large population of wintering cranes on the trophic status of Hula Valley waters. It seems publishable, but the authors need to make some revisions and answer some questions. The names of three lakes, Kinneret, Agmon, and Hula, scroll through the text - it would be helpful to illustrate this on a map.

the title of the article: Is Hula Valley and Lake Kinneret one ecosystem?  In scientific meaning – no. Better use : landscape

line 6 – old lake - that sounds awkward, maybe better to replace lakes undergoing terrestrialization

line 18 – km2 (and further down in the text)

Lines 120-121 The issue this paper: wants to solve is the reason for increase in TP concentration in the Agmon waters?. The aims of surveys should be rather presented in the Introduction.

line 138: isolated cranes – single cranes

lines 177-196 – Was this paragraph based on your research? If so, please include the results, if not, please complete the bibliographic sources.

line 198: [24] this paper provides information about only two bird species. Please expand the base add even works: 10.1111/j.1365-2427.2007.01838.x and 10.1007/s10750-015-2618-1

Lines 204 -205 : cranes and pelicans – 5.24 gP/bird/day; cormorants 3.5 gP/bird/day.( add the source)

Lines 230 -231 It is likely that most of the crane excretion takes place in the Agmon waters at night. - why you assume that?

line 242: re-suspension to resuspension

lines 289 290: These other sources are peat soil geochemical features and submerged aquatic plants (During the vegetation season submerged vegetation is sink for nutrients, so it cannot be responsible for the peak of phosphorus in water column, as you mentioned this may occur in autumn when the biomass start to decompose).

lines 326-336  I’m not sure if this paragraph is necessary, additionally the abbreviation for American currency is USD

Line 341 crane feeding block – what do you mean?

Reviewer 2 Report

Review of “Hydrology and Cranes (Grus grus) Attraction Partnership in the Management of Hula Valley – Lake Kinneret Ecosystem” by M. Gophen

The author is to be commended for analyzing the issue of cranes in the Hula Valley. There are many issues with English grammar that need to be resolved – these were not marked. The paper needs a lot of work to quantitatively assess the P balance and N balance in Lake Agmon and the Hula Valley. Many points were made – but none were a rigorous mass balance of P in this basin. Only this careful, quantitative approach can really answer the main research question.

Below are several comments to improve the quality of the paper.

Comments:

  1. Introduction: Need to also include the approximate elevation of the Hula valley
  2. Introduction: There absolutely needs to be a map showing the Hula Valley and the drainage basin of the Kinneret.
  3. Line 40-41: What does ‘enhancement of underground fires’ mean? Clarify.
  4. Line 44: Suggest delete: ‘The implementation resulted in partial modification to land utilization..’ It does not add anything to the paper.
  5. Line 44-50: A map is required to show these enhancements.
  6. Line 77-80: why capitalize ‘Reclamation Projects’ , ‘Nature’?
  7. Line 77-80: Needs rewritten – grammar.
  8. Line 82: This is redundant – this was just stated above in line 77.
  9. Line 99: Why has the subject moved to climate change? There was no discussion preceding this on that topic? What climate change? Rewrite so that this is clear.
  10. Line 99: ‘Information about climate change’ – what information? This is vague and needs to be rewritten.
  11. Fig 1 – poor quality reading the panels – make sure font is large enough to be read. The actual data must be shown – not just the polynomial regression. It is impossible to know how much data variability there is which is obscured by the regression lines.
  12. Line 104: This is usually just the opposite – why as Q increases TP increases?
  13. Fig caption consistency – Fig 2 and 3 – make sure you introduce each figure consistently.
  14. Fig 1,2,3: Why combine multiple figures into 1 figure – many of these are independent and do not to be grouped together.
  15. Figures – many of your figures only have ppm as a vertical scale – you must include what ppm represents, TP, Po4, TN, Org N, etc. It must be shown in the axis title not on the graph.
  16. The paper only gets to its objective on line 120-121. The paper should be rewritten with this as the focus if indeed that is the primary objective of this paper. From line 77-80 it seems like the objective is just a summary of information.
  17. 314: ‘possibility of P pollution …by cranes was rejected’ – this central question was not resolved. There was no careful mass balance of P in Lake Agmon showing what effect the P load has on the lake. This should be done by looking at storage and recycling of P quantitatively in the system – inflows, outflows, plant biomass accumulation and decay, nutrient addition from birds, agricultural runoff, etc. This was not done in a quantitative way to reach the conclusion of the author.

Reviewer 3 Report

These studies summarize the different effects of the achievements of the recovery project in lakes Agmon and Kinneret from 1994 to 2018, it is an interesting study, however I have comments: The summary: it must be improved, where it makes clear the objective, method used, results, discussions and conclusions, all this must be appreciated in the summary. The introduction is good and punctual The materials and methods section should be improved, it should allow any reader to replicate the research, the way the data was collected would be important. Know in detail how the data was processed. I recommend that the results and discussions section be unified in the same section so that in this way they are more user-friendly with the reader and allow the results to be discussed as they are presented in the manuscript, it would be valuable to have this comment found and restructured. Finally, I consider that the manuscript provides valuable information in decision-making, for which I consider that if these recommendations are attended to, it can be published in this journal.